# Clinical and Analytical Performance of ELISA Salivary Serologic Assay to Detect SARS-CoV-2 IgG in Children and Adults

**DOI:** 10.3390/antib13010006

**Published:** 2024-01-05

**Authors:** Andrea Padoan, Chiara Cosma, Costanza Di Chiara, Giulia Furlan, Stefano Gastaldo, Ilaria Talli, Daniele Donà, Daniela Basso, Carlo Giaquinto, Mario Plebani

**Affiliations:** 1Department of Medicine (DIMED), University of Padova, 35128 Padova, Italy; andrea.padoan@unipd.it (A.P.); chiara.cosma@unipd.it (C.C.); daniela.basso@unipd.it (D.B.); mario.plebani@unipd.it (M.P.); 2UOC of Laboratory Medicine, University-Hospital of Padova, 35128 Padova, Italy; 3QI.LAB.MED, Spin-off of the University of Padova, 35011 Padova, Italy; giulia.furlan.5@studenti.unipd.it; 4Department of Women’s and Children’s Health, University of Padova, 35128 Padova, Italystefano.gastaldo@studenti.unipd.it (S.G.); daniele.dona@unipd.it (D.D.); carlo.giaquinto@unipd.it (C.G.); 5Penta–Child Health Research, 35127 Padua, Italy

**Keywords:** SARS-CoV-2 antibodies, ELISA, salivary samples, children, adults

## Abstract

Saliva is a promising matrix with several purposes. Our aim is to verify if salivary anti-SARS-CoV-2 antibody determination is suitable for monitoring immune responses. One hundred eighty-seven subjects were enrolled at University-Hospital Padova: 105 females (56.1%) and 82 males (43.9%), 95 (50.8%) children and 92 (49.2%) adults. Subjects self-collected saliva using Salivette; nineteen subjects collected three different samples within the day. A serum sample was obtained for all individuals. The N/S anti-SARS-CoV-2 salivary IgG (sal-IgG) and serum anti-SARS-CoV-2 S-RBD IgG (ser-IgG) were used for determining anti-SARS-CoV-2 antibodies. The mean (min–max) age was 9.0 (1–18) for children and 42.5 (20–61) for adults. Of 187 samples, 63 were negative for sal-IgG (33.7%), while 7 were negative for ser-IgG (3.7%). Spearman’s correlation was 0.56 (*p* < 0.001). Sal-IgG and ser-IgG levels were correlated with age but not with gender, comorbidities, prolonged therapy, previous SARS-CoV-2 infection, or time from last COVID-19 infection/vaccination. The repeatability ranged from 23.8% (7.4 kAU/L) to 4.0% (3.77 kAU/L). The linearity of the assay was missed in 4/6 samples. No significant intrasubject differences were observed in sal-IgG across samples collected at different time points. Sal-IgG has good agreement with ser-IgG. Noninvasive saliva collection represents an alternative method for antibody measurement, especially in children.

## 1. Introduction

The World Health Organization (WHO) declared the end of the coronavirus disease 2019 (COVID-19) as a public health emergency on 5 May 2023 [1]. Nevertheless, there is no doubt that the COVID-19 pandemic between 2019 and 2022 resulted in high morbidity and mortality on a global scale and that community-level immunity, either acquired through SARS-CoV-2 infection or vaccination, played a crucial role in controlling the pandemic.

At present, analyzing global immunity is still of interest in order to clearly understand how and who to test for protecting fragile patients, such as patients with immunosuppressive status [2]. Indeed, mRNA COVID-19 vaccines have been demonstrated to induce a strong cellular [3] and humoral [4] response to SARS-CoV-2, with a progressive decline observed over six months post vaccination, regardless of previous COVID-19 disease. However, it has been documented that in some groups of patients, immunity is not effectively stimulated, neither by vaccines nor direct viral infection, leading to a possible reinfection of these patients [5]. Moreover, it has been shown that circulating antibodies against SARS-CoV-2 can persist for up to 18 months, but it is not clear whether there could be a decline in immunological memory, especially in asymptomatic infected individuals [6]. Therefore, the measurement of antibody titers developed against SARS-CoV-2 infection or that are vaccine-induced can be useful to facilitate the understanding of community-level immunity. The concomitant measurement of T-cell immunity has been helpful in defining individuals with a poor immunological response and in identifying previously asymptomatic infected children in case of suspicious multisystemic inflammatory syndrome (MIS-C) [7].

To facilitate large-scale serosurveillance, it is essential to employ robust and well-characterized assays that can be performed on easily accessible self-collected samples. In the last ten years, saliva has reached increasingly high importance for the evaluation of many aspects of human health [8]. The use of the salivary matrix to detect active SARS-CoV-2 infection is now well established both with the molecular real-time RT-PCR technique and with chemiluminescent enzyme immunoassays in highly automated platforms [9,10,11]: it is easy and noninvasive to collect and fast and cost-effective to analyze, allowing for widespread execution of the tests. Moreover, it is suitable for large-scale serial sampling for epidemiological studies and screening tests, since it may be well-accepted as a sampling method also by fragile, geriatric, and pediatric patients [12,13]. Furthermore, the opportunity to measure both antibodies and viral RNA in one single specimen makes saliva a valuable specimen to monitor individual and population SARS-CoV-2 transmission, infection, and seropositivity [14], in addition to being used for molecular diagnosis for viral RNA detection.

Since SARS-CoV-2 is a respiratory virus, the detection of antibodies at the sites of primary viral infection needs further investigation [15]. In this study, we aimed to investigate and validate the clinical, preanalytical, and analytical performances of an enzyme-linked immunosorbent assay (ELISA) to detect IgG antibodies against SARS-CoV-2 in saliva. In order to test the analytical performance of the assay, both salivary and serum samples of adult and pediatric patients who were infected and/or received COVID-19 mRNA vaccination were evaluated.

## 2. Materials and Methods

### 2.1. Study Design and Sample Collection

We conducted a single-center, observational study on pediatric patients (age ≤ 18 years) and adults who attended the COVID-19 Family Cluster Follow-up (CASE cohort) at the Department of Women’s and Children’s Health, University Hospital of Padova enrolled from January to October 2022. Parents or legally authorized representatives were informed of the research proposal and provided their written informed consent. In addition, a cohort of healthcare workers (HCWs) were included.

SARS-CoV-2 positivity among individuals, including both children and adults within the CASE cohort, was identified through molecular rRT-PCR testing performed on the viral genome extracted from nasopharyngeal swabs. Upon enrollment, a pediatrician (Costanza Di Chiara and Daniele Donà) gathered demographic data, medical history, SARS-CoV-2 molecular rRT-PCR test results, and vaccination status of both children and their parents. Additionally, a clinical evaluation was conducted.

For participants in the HCW cohort, routine screenings were conducted to detect SARS-CoV-2 infection every 2 or 3 weeks through molecular rRT-PCR testing.

All included subjects were educated in collecting saliva samples using Salivette (Sarstedt, Germany). In detail, the procedure for saliva collection included avoiding eating and drinking, performing normal oral hygiene at least 1 h before collection, and keeping the swab contained inside the Salivette device in the mouth for 1 min. On the same day of saliva collection, all subjects underwent blood sampling for SARS-CoV-2 S-RBD IgG serological test. Salivary samples were centrifuged at 4000 rpm for 5 min at room temperature (RT) and then frozen at −20 °C until use. Blood contamination of saliva was excluded by visual assessment of all samples after centrifugation. Blood samples were left clotting for 30 min at RT, centrifuged at 4000 rpm for 5 min, and then frozen at −80 °C until use.

### 2.2. Samples Analysis

The N/S anti-SARS-CoV-2 salivary IgG (ELISA) (RayBiotech Parkway Lane Suite, Peachtree Corners, GA, USA) assay (anti-SARS-CoV-2 N/S IgG) was used. This assay allowed us to quantitatively determine IgG against the nucleocapsid protein (N) and receptor-binding domain (RBD) (part of the S1 subunit of the spike protein). This method uses a plate coated with the SARS-CoV-2 N and S1-RBD proteins, which combine with the antibodies present in the sample. After one hour of incubation, the plate is washed and biotinylated IgG antibody is added to each well. This is followed by a short incubation of 30 min followed by a series of washings and the addition of the horseradish peroxidase (HRP)-streptavidin solution. After 30 more minutes of incubation and 5 washes, the TMB (3,3′,5,5′-tetramethylbenzidine) substrate solution is added and finally, after a 15 min incubation, the solution is added acid to stop the reaction. The same procedure is performed on another plate coated with human albumin which is used as a blank. The results from the albumin-coated plate should be subtracted from those obtained from the N/S1-RBD SARS-CoV-2 IgG protein-coated plate. No cross-reactivity data were available from the manufacturer.

Serum SARS-CoV-2 S-RBD IgG was performed using an already validated assay [16] (with a sensitivity of 91.7% (95%CI: 73.0–99.0), specificity of 91.8% (95%CI: 86.6–95.5) for mild symptomatic or asymptomatic patients, repeatability from 5.32% to 3.98%, and intermediate precision from 12.2% to 6.9%) by an automated platform Maglumi 2000 Plus (Snibe Co., Ltd., Shenzhen, China), which exploits the principle of chemiluminescence with paramagnetic particles coated with recombinant S-RBD antigen.

### 2.3. Precision Assessment

Precision was evaluated by using 6 saliva samples with different levels of anti-SARS-CoV-2 N/S IgG. For 5 of these samples, 5 aliquots were prepared, pipetted in random wells in one plate, and then analyzed. The last sample, with a mean concentration of 2.01 kAU/L, was repeatedly analyzed 25 times and plated in consecutive wells of the same plate to achieve a robust estimation of repeatability.

### 2.4. Linearity Assessment

Linearity was assessed by using a series of six sample pools, prepared with different anti-SARS-CoV-2 N/S IgG values, as specified in the CLSI EP06 A: 2003 guideline (paragraph 4.3.1) [17]. In brief, salivary pools were serially diluted with a pool of negative saliva with a value of anti-SARS-CoV-2 N/S IgG < 0.5 kAU/L. All measurements were performed in triplicate. A second-order polynomial regression was used to detect deviation from linearity.

### 2.5. Impact of Sample Collection Time on Salivary Ab Levels

In 19 subjects, salivary samples were collected using the same procedures specified above at three different times: (1) before breakfast or immediately after waking up; (2) during the morning between 10:00 and 11:00 a.m., and (3) after lunch, between 2:30 and 4:30 p.m. The reason for this choice was to test whether the time of testing had an impact on the analytical results of the assay as a preanalytical variable.

### 2.6. Statistical Analyses

Statistical analyses were performed using Stata v 16.1 (Stata Corp, Lakeway Drive, College Station, TX, USA). Descriptive statistics, the χ^2^ test, the Fisher exact test, and a 2-tailed, Kruskall–Wallis and unpaired *t*-test were used for categorical or continuous covariates. Linear regressions were used to assess the association between studied parameters and salivary or serum IgG after the transformation of the variables into base-10 logarithms. Friedman’s test for paired data was used to assess differences across the time of sample collection. Statistical significance was set at *p* < 0.05. All *p*-values were 2-sided. Graphs were made using GraphPad Prism, version 9.2 (GraphPad Software, Dotmatics, Boston, MA, USA).

### 2.7. Ethical Statement

The study protocol was approved by the local Ethics Committee (Prot. N°0070714 of 24 November 2020; amendment N°71779 of 26 November 2020).

## 3. Results

### 3.1. Performance Verification and Precision Analysis

The five repetitions of the first five samples had mean values ranging from 1.61 kUA/L to 10.1 kAU/L; precision (CV%) ranging from 23.6% (at a value of 7.4 kAU/L) to 3.9% (at a value of 3.77 kAU/L), and mean precision of 17.5% (SD± 8.5%) (Appendix A). In the last sample, with a mean concentration of 2.01 kAU/L, the 25 repetitions presented a mean CV of 11%.

### 3.2. Linearity Assessment

The six samples evaluated for linearity ranged from 20.1 kAU/L to 248 kAU/L (Figure 1). With the exception of the samples at values of 20.9 kAU/L (sample 1) and 27.9 kAU/L (sample 4), in which linearity was confirmed by polynomial regression; the other four samples demonstrated a marked absence of linearity.

### 3.3. Clinical Study

Table 1 summarizes the characteristics of the studied population.

A total of 95 (50.8%) pediatric patients and 92 (49.2%) adults were included in the study, with a ratio between females/males not significantly different for children (51.6%) and adults (60.8%). A small portion of the studied individuals (17.9% for pediatrics and 14.1% for adults) presented comorbidities, with ongoing prolonged therapy (17% for pediatrics and 10.1% for adults). Across major comorbidities, there were rheumatic diseases, inflammatory bowel diseases, renal diseases, chronic cephalea, and metabolic diseases (e.g., Hashimoto’s disease and hypothyroidism). Salivary anti-SARS-CoV-2 N/S IgG differed between pediatric and adults, being higher in the latter group (χ^2^ = 6.4, *p* = 0.0188, Table 1). A total of 58/95 (61.1%) and 66/92 (71.7%) of pediatric and adult patients, respectively, were positive for salivary anti-SARS-CoV-2 N/S IgG, and 83/95 (87.4%) and 88/92 (95.7%) pediatric and adults patients, respectively, were positive for serum anti-SARS-CoV-2 S-RBD IgG.

Table 2 reports the salivary anti-SARS-CoV-2 N/S IgG and serum S-RBD IgG titers in pediatric and adult patients at different time points from vaccine-induced immunization or from SARS-CoV-2 infection. Interestingly, salivary anti-SARS-CoV-2 N/S IgG titers differed between children and adults (*p* = 0.002) at >210 days after vaccine-induced immunization or SARS-CoV-2 infection.

Table 3 reports the univariate linear regression analyses of anti-SARS-CoV-2 N/S IgG with respect to all studied variables, subdivided by pediatric and adult individuals.

Considering pediatric subjects with and without previous SARS-CoV-2 infection, anti-SARS-CoV-2 N/S IgG median (and IQR) levels were similar, being 2.3 kAU/L (0.5–51.4 kAU/L) and 3.6 kAU/L (0.5–68.4 kAU/L) (*p* = 0.770), respectively. In adults, anti-SARS-CoV-2 N/S IgG median (and IQR) levels did not significantly differ between subjects with or without previous COVID-19 (29.5 kAU/L (0.5–92.1 kAU/L) and 51.8 kAU/L (0.5–137.2 kAU/L), *p* = 0.749). Anti-SARS-CoV-2 N/S IgG did not differ in subjects with or without comorbidities (χ^2^ = 2.98, *p* = 0.083 for children and χ^2^ = 2.94, *p* = 0.086 for adults) or in subjects with or without prolonged therapy (χ^2^ = 3.064, *p* = 0.080 for pediatric and χ^2^ = 0.042, *p* = 0.838 for adults) (Table 3).

Table 4 reports the univariate analyses of serum anti-SARS-CoV-2 S-RBD IgG with respect to all studied variables.

Serum anti-SARS-CoV-2 S-RBD IgG median levels (and IQR) significantly differ between subjects with or without previous COVID-19 in pediatric patients (306.2 kBAU/L (3.9–1554.9 kBAU/L) and 1523.7 kBAU/L (410.5–2739.2 kBAU/L), respectively, *p* < 0.001), but the difference is not statistically significant in adults (426.9 kBAU/L (297.5–4993.4) and 2171.3 (1101.3–4081.7), *p* = 0.121). The presence of comorbidities was significantly associated with anti-SARS-CoV-2 S-RBD IgG in pediatric patients and adults (Table 4). Prolonged therapy was related to the presence of antibodies in serum in children but not in adults (Table 4). The time from the last SARS-CoV-2 infection or vaccination was not associated with overall salivary anti-SARS-CoV-2 N/S IgG or with serum anti-SARS-CoV-2 S-RBD IgG, both in pediatric and adult patients (Table 3 and Table 4).

Table 5 reports multivariate analyses performed on salivary and serum of adult and pediatric samples for what concerns all the studied variables (the presence or absence of comorbidities, prolonged therapy, previous SARS-CoV-2 infection, and time from last SARS-CoV-2 infection or vaccination). Statistically significant differences were found in salivary and serum antibody titers for what concerns age in pediatric patients (*p* < 0.001 for both). However, none of the studied variables showed a statistically significant association with the presence of antibodies, with the exception of the presence or absence of comorbidities for serum anti-SARS-CoV-2 S-RBD IgG in adults.

### 3.4. Impact of Sample Collection Time on Salivary Ab

Figure 2 shows the intrasubject differences in anti-SARS-CoV-2 N/S IgG across all the collected samples. Although differences were observable, they were not statistically significant (Friedmans’ test *p* = 0.327).

## 4. Discussion

COVID-19 vaccination represents the standard care for preventing SARS-CoV-2 severe infection, and vaccines have been shown to elicit a strong cellular and immunological response, which protects against viral infection [3,18]. Thus, the assessment of SARS-CoV-2-specific antibodies is not typically sought, even if this analysis could be of relevance in evaluating the immunological status of fragile and immunocompromised patients [19]. Differently from adults, SARS-CoV-2 infection is notably mild in children [20]. However, in fragile children, such as immunosuppressed patients, transplant recipients, or with comorbidities [21] and/or prolonged therapies, COVID-19 might increase the risk of hospitalization [22,23]. In such situations, both in adults and children, identifying the existence of SARS-CoV-2 antibodies could hold clinical relevance; additionally, for diagnosis of multisystem inflammatory syndrome in children (MIS-C), serologic testing for SARS-CoV-2 is useful for the detection of past infection and thus for prescribing the correct therapy [24,25].

Rather than relying on serological tests to detect SARS-CoV-2 antibodies, salivary samples may be used as an alternative method [26,27]. This method could improve overall patient compliance, encompassing even those who are fragile, elderly, or children [28]. Saliva testing serves as a noninvasive source of antibodies for immunoassays, and it facilitates cost-effective epidemiological monitoring of infections [29]. However, antibody titers tend to be lower in saliva than in serum [30] and thus, assays should be carefully evaluated before use.

In the present study, an ELISA immunoassay was used to detect antibodies against S/N peptides of SARS-CoV-2 (anti-SARS-CoV-2 N/S IgG) in salivary samples; results were then compared with serum anti-SARS-CoV-2 S-RBD IgG through the use of an established chemiluminescence assay detecting the RBD portion of the viral spike protein [16]. Precision analysis of the assay demonstrated a mean CV of 17.5% across five salivary samples, each tested five times. One additional sample, with a mean concentration of 2.01 kAU/L was tested a total of 25 times, reporting a CV of 20.9%, further supporting the assay’s reliable repeatability, even at low anti-SARS-CoV-2 N/S IgG levels. The assay linearity was also studied, and because it can vary among different samples, six different specimens were tested. The ELISA assay lacked linearity for four out of the six samples analyzed, suggesting that it has some limitations in accurately quantifying anti-SARS-CoV-2 N/S IgG antibodies within certain concentration ranges. This fact should be considered relevant, especially if patients undergo antibody monitoring during this time [31].

For what concerns the clinical performances, some differences were found between saliva and serum samples. Firstly, the percentage of positivity to anti-SARS-CoV-2 N/S IgG was lower in children than in adults. These results are consistent with findings from Keuning et al. [32], who suggested that the elevated prevalence of positivity in serum compared with saliva could be partially explained by time kinetics and demographic differences. Secondly, adult patients reported higher SARS-CoV-2 antibody titers than children in both the salivary and serum samples. Differences between saliva and serum have been described by some studies [33,34], underlining that the persistence of elevated anti-SARS-CoV-2 antibodies in plasma may not indicate the persistence of antibodies at mucosal sites such as the nose or mouth [34]. Other authors demonstrated a high concordance between saliva and serological findings [14,29], while partial discordant results are reported by other studies [33], meaning that the salivary matrix still requires further investigation and validation as a source of biological analytes.

To better evaluate salivary and serum humoral responses, anti-SARS-CoV-2 titers were analyzed and broken down by time from vaccine-related immunization or SARS-CoV-2 infection. Differences in salivary anti-SARS-CoV-2 N/S IgG and serum S-RBD IgG titers between pediatric and adult patients were confirmed at <113, 113–180, 181–210, and >210 days from immunization. Moreover, there was a noticeable swift decrease over time in N/S-specific salivary IgG. While a decline of N/S-specific salivary IgG was also evident in adults, it occurred with a minor degree in children. Differently, for serum S-RBD-specific IgG, the time kinetic results indicate a gradual decline of equal magnitude in both children and adults (Table 2). Similar findings for time kinetics were reported by Keuning et al. [32], who studied salivary and serum N-specific, S-specific, and RBD-specific IgG in children and adults in a similar period of the pandemic wave.

Age-related factors may influence antibody titers both in children and in adults since age was found to be significantly associated with anti-SARS-CoV-2 antibody levels [35]. In addition, no statistically significant differences were found for anti-SARS-CoV-2 N/S IgG or for anti-SARS-CoV-2 S-RBD IgG between children and adults with and without previous SARS-CoV-2 infection at multivariate analyses (Table 4). These results are unexpected since a wane of Ab levels is well described in the literature [36,37,38]. This could be explained by the limited time from sample collection and vaccination or SARS-CoV-2 infection. Lastly, even though small antibody variations were found depending on the collection time of saliva, they were not statistically significant in either patient, suggesting the possibility of collecting samples at different time points during the day, including outpatient visits.

Saliva has already been tested by other authors, demonstrating a high concordance with serological findings in some studies [14,29] and partial discordant results in other cases [33], meaning that the salivary matrix still requires further investigation and validation as a source of biological analytes. Although the humoral response to SARS-CoV-2 by specific antibodies has been extensively investigated in serum samples of SARS-CoV-2-infected patients and vaccinated patients [39,40], local humoral immunity in the oral cavity and its relationship to systemic antibody levels need to be further addressed [41].

This study presents several limitations, such as a small sample size and a large variability in time from the last SARS-CoV-2 infection or last COVID-19 vaccination. Another limitation is the absence of measurement of salivary IgA, which is the predominant immunoglobulin present in saliva. Further, blood contamination of saliva was not tested by measuring the hemoglobin levels, and cross-reactivity with other human coronaviruses was not evaluated. On the other hand, a strength of the study is represented by the well-characterized cohort of pediatric patients.

## 5. Conclusions

In conclusion, mucosal immunity could provide valuable data that are useful to deepen the understanding of the SARS-CoV-2 immune response and antibody presence [42]. The use of alternative matrices found at the site of viral entry (i.e., oral cavity) may provide further information and novel analytical methods for antibody testing in the general population in order to monitor immunity induced either by previous SARS-CoV-2 infection or vaccination, especially for fragile patients.

## Figures and Tables

**Figure 1 antibodies-13-00006-f001:**
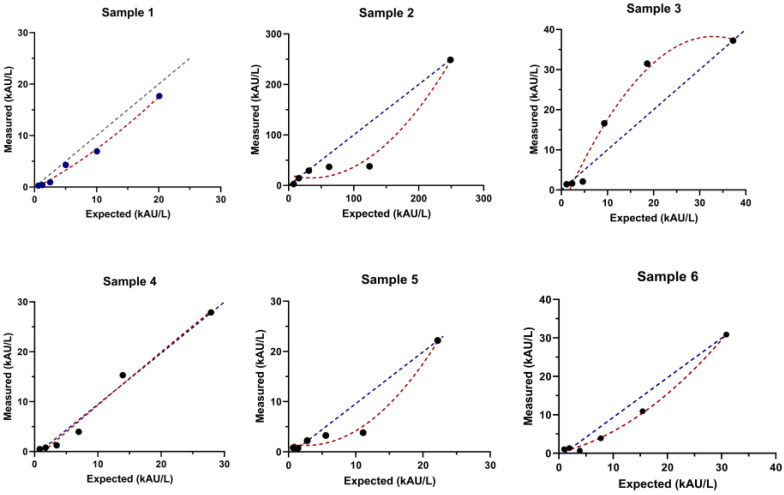
Linearity assessment for six samples. Linearity was assessed by polynomial regression. Dots are the concentration obtained from dilution. The blue color line is the dilution expected and the red color line is the dilution obtained.

**Figure 2 antibodies-13-00006-f002:**
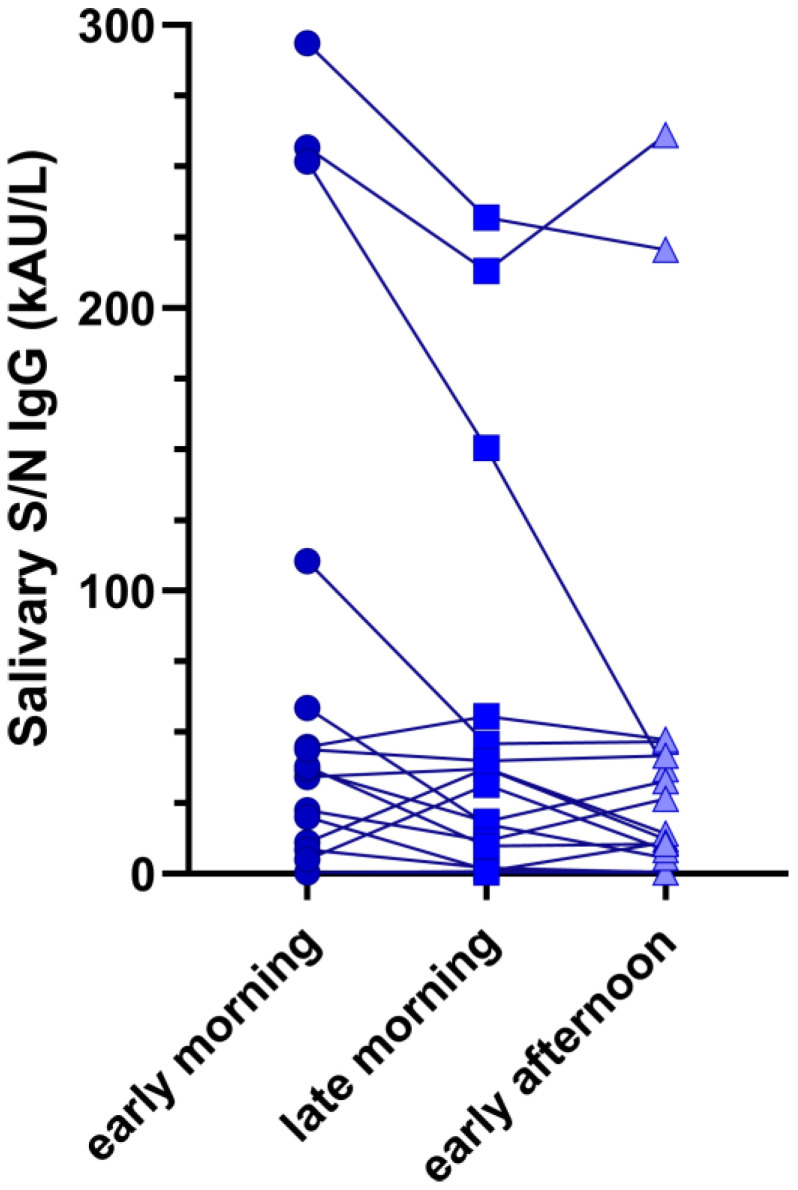
Intrasubject differences in anti-SARS-CoV-2 N/S IgG across all collected samples. The circle dots represent the concentration of in anti-SARS-CoV-2 N/S IgG in the early morning while the square and triangular points the concentrations in the late morning and early afternoon respectively.

**Table 1 antibodies-13-00006-t001:** Demographics and clinical characteristics of the subjects included in the study.

	Overall	Children	Adults	*p*-Value *
** *n* **	187	95 (50.8%)	92 (49.2%)	-
**Age (mean ± SD)**	24.66 ± 17.15	9.01 ± 3.3	40.8 ± 8.4	*t* = 34.4, *p* < 0.001
**Gender**	105/82 (F/M)	49/46 (F/M)	56/36 (F/M)	Fisher’s exact test, *p* = 0.239
**Comorbidities**	30/187 (16.0%)	17/95 (17.9%)	13/92 (14.1%)	Fisher’s exact test, *p* = 0.552
**Prolonged therapy**	25/183 ^	16/94 (17.0%) ^	9/89 (10.1%) ^	Fisher’s exact test, *p* = 0.201
**Previous SARS-CoV-2 infection**	154/187 (82.35%)	80/95 (84.21%)	74/92 (80.43%)	Fisher’s exact test, *p* = 0.567
**Time from last COVID-19 or vaccination (median and IQR ^$^)**	175.3 ± 82.7	174.5 ± 72.5	176.1 ± 92.3	t = 0.119, *p* = 0.904
**Salivary N/S RBD IgG (kAU/L)**	29.7 (0.5–87.4)	2.96 (0.5–67.5)	50.6 (0.5–127.6)	χ^2^ = 6.4, *p* = 0.019
**Anti SARS-CoV-2 S-RBD IgG (kBAU/L) (median and IQR ^$^)**	1672.5 (514.4–3345.8)	1138.8 (321.3–2706.3)	2022.9 (808.4–4118.3)	χ^2^ = 10.4, *p* = 0.001

* significance between pediatric and adult patients; $ interquartile range (25th and 75th percentile); ^ prolonged therapy data were available for a total of 183 subjects.

**Table 2 antibodies-13-00006-t002:** Salivary anti-SARS-CoV-2 N/S IgG and serum anti-SARS-CoV-2 S-RBD IgG titers in pediatric and adult patients at different time points from vaccine-induced immunization or from SARS-CoV-2 infection. Median and interquartile ranges (25th to 75th percentiles) are reported.

Time from Vaccine-Induced Immunization or Time from SARS-CoV-2 Infection	Salivary N/S IgG (kAU/L), Median (IQR), (*n*)		Serum S-RBD IgG (kBAU/L) Median (IQR), (*n*)	
	Children	Adults	Children	Adults
**<113 days**	2.01 (0.5 to 18.1) (*n* = 21)	59.4 (2.02 to 117.9) (*n* = 22)	1719 (526 to 3596) (*n* = 21)	3136 (1382 to 4178) (*n* = 22)
**113 to 180 days**	12.2 (0.5 to 60.7) (*n* = 16)	73.8 (0.5 to 192.0) (*n* = 23)	1302 (431 to 2521) (*n* = 16)	2319 (419 to 5364) (*n* = 23)
**181 to 210 days**	31.9 (0.5 to 110.2) (*n* = 28)	47.5 (0.5 to 68.8) (*n* = 17)	1426 (378 to 2821) (*n* = 28)	1961 (787 to 3345) (*n* = 17)
**>210 days**	2.2 (0.5 to 47.1) (*n* = 30)	49.1 (1.3 to 328.5) (*n* = 30)	800 (89 to 2340) (*n* = 30)	1585 (800 to 5523) (*n* = 30)

**Table 3 antibodies-13-00006-t003:** Univariate analyses for log10 Salivary N/S RBD IgG (kAU/L).

	Children		Adults	
Variables	Coefficients (95%CI)	*p*-Value	Coefficients (95%CI)	*p*-Value
**Age**	0.13 (0.07–0.20)	*p* < 0.001	−0.03 (−0.59–(−0.001))	*p* = 0.040
**Gender**	0.09 (−0.37–0.55)	*p* = 0.686	0.13 (−0.37–0.63)	*p* = 0.600
**Comorbidities**	−0.50 (−1.08–0.09)	*p* = 0.095	−0.57 (−1.27–0.13)	*p* = 0.110
**Prolonged therapy**	−0.57 (−1.12–0.09)	*p* = 0.092	−0.06 (−0.89–0.78)	*p* = 0.890
**Previous SARS-CoV-2 infection**	0.10 (−0.58–0.76)	*p* = 0.770	−0.52 (−1.12–0.09)	*p* = 0.750
**Time from last COVID-19 or vaccination**	−0.001 (−0.004–0.001)	*p* = 0.316	0.0001 (−0.003–0.004)	*p* = 0.730

**Table 4 antibodies-13-00006-t004:** Univariate analyses for log10 anti-SARS-CoV-2 S-RBD IgG (kBAU/L).

	Children		Adults	
Variables	Coefficients (95%CI)	*p*-Value	Coefficients (95%CI)	*p*-Value
**Age**	0.08 (0.20–0.15)	*p* = 0.010	−0.01 (-0.23–0.01)	*p* = 0.550
**Gender**	0.04 (−0.35–0.44)	*p* = 0.830	0.04 (−0.26–(0.35))	*p* = 0.780
**Comorbidities**	−0.86 (−1.34–(−0.38))	*p* = 0.001	−0.61 (−1.01–(−0.2))	*p* = 0.004
**Prolonged therapy**	−0.95 (−1.43–(−0.48))	*p* < 0.001	−0.33 (−0.83–0.16)	*p* = 0.190
**Previous SARS-CoV-2 infection**	0.95 (0.43–1.47)	*p* = 0.001	0.29 (−0.08–0.67)	*p* = 0.120
**Time from last COVID-19 or vaccination**	0.0001 (−0.002–0.003)	*p* = 0.910	−0.001 (−0.003–0.0003)	*p* = 0.110

**Table 5 antibodies-13-00006-t005:** Multivariate analysis for log10 salivary N/S IgG (kAU/L) and for log10 anti-SARS-CoV-2 S-RBD IgG (kBAU/L).

	log10 Salivary Anti-SARS-CoV-2 N/S IgG		log10 Anti-SARS-CoV-2 S-RBD IgG	
	Children	Adults	Children	Adults
	Coefficients (95%CI), *p*-Value	Coefficients (95%CI), *p*-Value	Coefficients (95%CI), *p*-Value	Coefficients (95%CI), *p*-Value
**Age**	33.9 (14.5 to 53.3), *p* = 0.001	−2.99 (−13.30 to 7.31), *p* = 0.565	0.093 (0.044 to 0.142), *p* < 0.001	−0.002 (−0.019 to 0.015), *p* = 0.785
**Gender**	134.9 (5.6 to 264.1), *p* = 0.041	176.4 (−2.9 to 355.6), *p* = 0.054	0.22 (−0.09 to 0.54), *p* =1.44	0.16 (−0.12 to 0.45), *p* = 0.266
**Comorbidities**	219.9 (−56.9 to 496.7), *p* = 0.158	−153.5 (−524.3 to 217.4), *p* = 0.412	0.355 (−0.351 to 1.062), *p* = 0.320	−0.93 (−1.53 to −0.34), *p* = 0.002
**Prolonged therapy**	−45.7 (−327.2 to 235.7), *p* = 0.747	226.3 (−191.9 to 644.5), *p* = 0.285	−0.43 (−1.12 to 0.26), *p* = 0.218	0.42 (−0.25 to 1.09), *p* = 0.216
**Previous SARS-CoV-2 infection**	130.9 (−89.6 to 351.6), *p* = 0.152	−68.7 (−295.9 to 158.4), *p* = 0.548	0.21 (−0.31 to 0.73), *p* = 0.429	0.11 (−0.25 to 0.48), *p* = 0.533
**Time from last COVID-19 or vaccination (days)**	−0.82 (−1.71 to 0.07), *p* = 0.070	−0.099 (−1.073 to 0.874), *p* = 0.839	0.0002 (−0.002 to 0.002), *p* = 0.856	−0.001 (−0.003 to 0.0003), *p* = 0.109

## Data Availability

Data are contained within the article and supplementary materials.

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
