# Peer review of "Clinical and Analytical Performance of ELISA Salivary Serologic Assay to Detect SARS-CoV-2 IgG in Children and Adults"

_2073-4468, 2024, doi:10.3390/antib13010006_

Round 1

Reviewer 1 Report

Comments and Suggestions for Authors

Summary

This paper aims to evaluate the relationship between the anti-SARS-CoV-2 antibody in saliva and the immune response which determined by serum anti-SARS-coV-2 S-RBD IgG. The authors collected samples from 187 subjects that were enrolled in University-Hospital Padova. The results showing that salivary IgG correlate serum IgG with age but other factors including gender, comorbidities, prolonged therapy, previous SARS-CoV-2 infection and infection or vaccination time.

General concept comments:

Article:

In general, this article is well-written and clearly shown the statistical analysis for each of the potential factor that can be correlated with saliva anti-SARS-CoV-2 antibodies with well-specified statistical method. The discussion part and conclusion are reasonable from the results or observations. There is no major issue that can be found in this article. It will be more compelling if adding more factors.

Author Response

Thanks very much for appreciating our work.

Reviewer 2 Report

Comments and Suggestions for Authors

General Comments:

The   paper   “Clinical and Analytical Performance of ELISA Salivary Sero-2 logic Assay to Detect SARS-CoV-2 IgG in Children and Adults”   by Padoan   et  al   seeks to investigate the clinical, pre-analytical, and analytical performances of an enzyme-linked im-76 immunosorbent assay (ELISA) to detect IgG antibodies against SARS-CoV-2 in saliva. In order to test the analytical performance of the assay, both salivary and serum samples of adult and pediatric patients were evaluated, who were infected and/or received COVID-79 19 mRNA vaccination.

In general, the  paper  could have potential value in  providing  a  useful  predictive  marker of  protective immunity to SARS-CoV-2 infection but there are a number of deficiencies in the design of the study that preclude the authors from achieving their  goal.  

Specific  Comments

Although the authors seem  to  recognize  (bbut  never  directly state) that salivary antibody could be generated either by infection or immunization with mRNA vaccines they do not include any  detailed information on the timing of immunization or  the  timing of  COVID-19  infection. So it makes the interpretation of the   biologic significance of their salivary IgG findings difficult   or impossible to make.  A  Table  describing  these timing events would  solve  this  problem.   The  authors  would  benefit  from     reading  and referring  to  inn  their article  the  paper  by  Li D, et  al. Salivary and serum IgA and IgG responses to SARS-CoV-2-spike protein following SARS-CoV-2 infection and after immunization with COVID-19 vaccines. Allergy Asthma Proc. 2022 Sep 1;43(5):419-430  in  which  these  timing   events  were   very  well  documented.

Also, since   the paper is describing events concerned with mucosal  immunity,  the  authors  fail  to  measure   or  even  mention    the  predominant  immunoglobulin in    saliva  and  respiratory  secretions namely  the secretory IgA immunoglobulins.  This  a  serious  deficiency.  

There  are also a  number of deficiencies in referencing misstatements  throughout  the paper .  For  example.on  page 2 the  statement ‘In the last ten years, saliva reached increasingly high importance for the evaluation of many aspects of human health’ needs  to be  referenced with  several  references.   The statement   ‘Blood contamination of saliva was 104 excluded by visual assessment of all samples after centrifugation’ on  page 3 is hardly the way to exclude contamination of salivary specimens with blood. This should have been done with more sensitive assays such as hematest.

Finally it is not clear why several specimens  were taken during the  course  of  a  day  and why linearity assessment for six samples. Linearity was assessed by polinomial regression was  performed  and  these  limited measurements mean.

Comments on the Quality of English Language

  Needs  improvement

Reviewer 3 Report

Comments and Suggestions for Authors

1. Al line-90, the authors used some abbreviation (CDC, DD) without expressing any full word at the manuscript. Please revise it.

2.The authors described that participants were enrolled after checking RT-PCr. Which type of specimens are using to detect viral genome. The authors should describe full information either nasopharyngeal swab or saliva? Pleas add at the revised manuscript

3. The authors described that ELISA system was already validated. Please add the sensitivity and specificity of your ELISA system. How about the cross reactivity of IgG  Ab with other corona viruses and closely related viruses 

4. Please briefly discuss about the cross reactivities of your tested IgG Ab with other corona viruses.

5. Among the study participants, how many participants were positive for ELISA for saliva samples and serum samples. This is very important and please add it.

Comments on the Quality of English Language

Please check language again with native speakers.

Round 2

Reviewer 2 Report

Comments and Suggestions for Authors

Authors have responded adequality to reviewer suggestions

Reviewer 3 Report

Comments and Suggestions for Authors

Thank you very much for revising all comments.